# Blood transfusion service readiness and its associated factors in health facilities providing blood transfusion services across Ethiopia: A secondary analysis of the 2018 Service Availability and Readiness Assessment (SARA) survey

**Tirualem Asmare Fenta[1], Gizachew Tadele Tiruneh[2]\*, Nebiyou Fasil Ayele[3]**

1 Ethiopian Food and Drug Administration, Addis Ababa, Ethiopia, 2 JSI Research & Training Institute, Inc. and Addis Continental Institute of Public Health, Addis Ababa, Ethiopia, 3 Addis Continental Institute of Public Health, Addis Ababa, Ethiopia

\* gizt121@gmail.com

**Data Availability Statement:** Conducted by the Ethiopian Public Health Institute (EPHI) between

## Abstract

### Introduction

Timely and safe blood transfusion services are crucial for saving lives in emergencies. Previous studies have focused on hospital inpatient care access but have overlooked blood transfusion service readiness. This study examined the readiness of blood transfusion services in health facilities across Ethiopia and its determinants.

### Methods

This study used Service Availability and Readiness Assessment (SARA) 2018 data from 632 facilities. Readiness was measured based on seven components: the presence of at least one trained staff for appropriate use of blood and safe transfusion; guidelines for appropriate use and transfusion; blood storage refrigerator; blood typing; cross-match typing; blood supply safety; and blood supply sufficiency. Data were analyzed using descriptive and inferential statistics. Facility characteristics were summarized using frequency tables and summary statistics. Bivariate and multivariable linear regression analyses examined the predictors of service readiness.

### Results

Facilities offering blood transfusion services had a mean readiness score of 4.5 (out of 7), with only 5% having all items. Most facilities performed blood typing, but less than one-third conducted cross-match testing, and over half lacked guidelines and trained staff. Service readiness varied significantly across regions. Facilities in Oromia (Coef.: -0.74; 95% CI: [-1.32, -0.15]) and Somali (Coef.: -1.26; 95% CI: [-2.21, -0.31]) had lower readiness scores

October and December 2017, the SARA 2018 dataset—generated from a facility-based cross-sectional survey—offers insights into service availability and readiness for general healthcare and various specific services at both national and regional levels. Interested researchers can access the dataset by following a data-sharing agreement. Researchers can request the dataset through EPHI's data repository website for the "Ethiopia Services Availability and Readiness Assessment (SARA) Survey, 2018," available at https://rtds.ephi.gov.et/public/showdetail/94. An assigned officer will review and facilitate the request. Upon submission, researchers will receive an email outlining the requirements, which include: 1) completing and returning the data request form; 2) providing a copy of the ethical approval certificate; and 3) attaching an official data request letter.

**Funding:** The author(s) received no specific funding for this work.

**Competing interests:** The authors have declared that no competing interests exist.

**Abbreviations:** ACIPH, Addis Continental Institute of Public Health; ANOVA, one-way analysis of variance; AOR, adjusted odd ratio; CI, confidence interval; EmONC, Emergency Obstetric and Newborn Care; EPHI, Ethiopian Public Health Institute; LMICs, low-and middle-income countries; NBBS, National Blood Bank Services; OR, odds ratio; SARA, Service Availability and Readiness Assessment; SNNP, Southern Nations, Nationalities, and Peoples' region; SPA, Service Provision Assessment; VIF, variance inflation factor; WHO, World Health Organization.

compared to Addis Ababa. Increased availability of medical equipment corresponded to a 49% increase in readiness scores (Coef.: 0.49; 95% CI: [0.19, 0.79]).

## Conclusion

The study highlights deficiencies in blood transfusion service readiness and regional disparities, emphasizing the need for targeted support to enhance readiness across regions.

## Introduction

Injuries, violence, and obstetric hemorrhage present critical health challenges worldwide, especially in low- and middle-income countries (LMICs) [1]. Each year, approximately 4.4 million people die globally due to injuries and violence, with obstetric hemorrhage accounting for about 0.3 million maternal deaths [2]. Safe blood transfusion is essential for treating injuries and emergency obstetric conditions, serving as a life-saving intervention in various clinical scenarios. Timely and secure blood transfusion services are particularly vital in LMICs, where over half of deaths result from conditions manageable through emergency care [3, 4]. Ethiopia, a developing country, faces a chronic shortage of blood, contributing to high maternal mortality rates, with 30% of maternal deaths due to hemorrhage resulting from the unavailability or delays in blood transfusion services [5].

However, LMICs face challenges in providing safe blood and transfusion services despite collecting 60% of global blood donations, as they are home to 84% of the world's population [6, 7]. In Ethiopia, less than half of the required blood units are collected annually, though the country needs more than 300,000 units per year [8]. Evidence shows significant challenges facing the sub-Saharan Africa region in ensuring a safe blood supply, including inadequate infrastructure, limited financial resources, and a high prevalence of bloodborne infections [9].

Health facilities may fail to supply safe blood for critical patients due to inadequate readiness [10, 11]. Blood transfusion service readiness—defined as the preparedness and capability of a facility to provide safe, timely, and effective blood products and services, which includes trained staff, blood storage refrigerators, blood typing, cross-matching, supply safety and sufficiency, and guidelines for appropriate use—is crucial for ensuring safe and effective transfusions [12]. Several factors can influence facility readiness, including infrastructure, equipment, human resources, supply chain management, monitoring, and quality assurance [9, 13–15].

Previous studies have largely focused on access to broad hospital inpatient care for severe conditions but have not accounted for the readiness of blood transfusion services. There is limited evidence on blood transfusion service readiness in facilities offering blood transfusion and its determinants across Ethiopia. This study aims to examine the service readiness in facilities offering blood transfusion and its determinants in Ethiopia, providing policymakers and program managers with evidence-based decisions for quality services.

## Methods

### Settings

This study is based on secondary national SARA 2018 survey data conducted by the Ethiopian Public Health Institute (EPHI) [16]. The SARA survey was conducted at selected health facilities in all regions of the country to assess and monitor the service availability and readiness of

the health sector and to generate evidence to support the planning and managing of a health system [16].

Ethiopia is administratively divided into four levels: regions, zones, woredas (districts), and kebeles. Before the survey, the country was divided into 9 regional states and 2 chartered cities: Tigray, Afar, Amhara, Oromia, Somali, Benishangul Gumuz, Sothern Nations, Nationalities, and Peoples' region (SNNP), Gambella, Harari, Dire Dawa, and Addis Ababa. After 2021, SNNP is further divided into 4 regions namely Central Ethiopia, South Ethiopia, Sidama, and South West Ethiopia. As such, currently, the country is divided into 12 regional states and 2 town administrations.

Ethiopia's health system has three tier levels: primary level care (encompassing a primary hospital, health centers, and health posts), secondary level care, and tertiary level care [17]. In 2020, 367 public hospitals, 47 private hospitals, 3,777 health centers, 3,867 private clinics, and 17,699 health posts were providing health services to the population [18]. Hospitals provide maternal and child health services (including prenatal care, safe delivery, and postpartum care), emergency medical services for injuries and critical medical conditions, inpatient for intensive and surgical care, and outpatient consultations, vaccinations, and follow-up care. While health centers provide preventive and curative maternal and child health services and outpatient services including referral care.

Blood transfusion services are provided by the Ethiopian Red Cross Society and government health facilities, with the National Blood Bank Services (NBBS) now operating under the Ministry of Health. The NBBS oversees 25 blood banks, including a central one in Addis Ababa and 24 regional banks, ensuring access to safe blood across the country. These blood banks cover the requirements of 52% of hospitals in Ethiopia [19].

## Data

SARA survey employed a cross-sectional facility-based design that was carried out in all regions of the country. The sampling method employed was a nationally representative sample stratified by health facility type and managing authority (public and private facilities). Accordingly, accounting for the skewed health facility distribution in the regions, the SARA 2018 survey enumerated 764 facilities, a census of 303 hospitals, 164 health centers, 165 clinics, and 132 health posts. For this study, data from 632 facilities (excluding health posts that not are expected to provide blood transfusion services) was used. Two hundred fifty-six (41%) of them offered blood transfusion services. The adequacy of the sample size was confirmed, with the available 256 facilities offering blood transfusion services being sufficient to address the study objectives. The adequacy of the sample size is assessed using a single proportion formula with a 5% margin of error, a 95% confidence level, a design effect of 1.5, and an assumption that 88% of hospitals reported performing compatibility testing for whole blood and RBC [20]. The Wilson score interval method was used to generate the confidence interval for the proportion estimate. Additionally, a double population comparison is conducted for blood transfusion service readiness, considering managing authority and facility type as exposure variables, with an assumed power of 80%, a design effect of 1.5, and a 95% confidence interval [16].

Data were collected from October—December 2017 using pretested facility inventory standard questionnaires translated into Amharic. Eighty-nine trained data collectors and seven regional coordinators participated in the data collection. Tablet computers, a computer-assisted personal interviewing technique, were used during interviews to ask questions and record responses. A facility inventory questionnaire was used to collect information on the availability of specific items (including their location and functional status), components of support systems, facility infrastructure, including the service delivery environment, and service

availability. The availability of diagnostic, essential medicines, and infrastructure resources, and the readiness of health facilities to provide basic health care interventions relating to maternal health, child health services, HIV/AIDS, tuberculosis, malaria, and non-communicable diseases were assessed. Specific to facility readiness, the interviewer was instructed to interview the person most knowledgeable about blood transfusion services at the facility where blood is collected, processed, tested, stored, or handled before transfusion.

For this study, the researcher obtained the full dataset from EPHI after signing the data-sharing agreement form on January 15, 2024. From the dataset, information related to facility characteristics (location, managing authority, facility type, support systems, facility infrastructure, including the service delivery environment, etc.) and service readiness in facilities providing blood transfusion services were extracted.

## Measurement

**Blood transfusion service readiness in health facilities providing blood transfusion.**
Blood transfusion service readiness was assessed based on the availability of essential medicines, trained personnel, medical equipment, and guidelines, following the WHO Service Availability and Readiness Assessment (SARA) guide [12]. Health facilities offering blood transfusion services were assessed on service readiness based on the availability of the 7 tracer items that include the presence of at least one trained staff for appropriate use of blood and safe blood transfusion, guidelines for appropriate use of blood and blood transfusion, blood storage refrigerator, blood typing, cross match typing, blood supply safety, and blood supply sufficiency. Table 1 below presents the items included for measuring readiness and their definitions.

The mean percentage service readiness score was calculated based on the above-mentioned tracer items by coding available as "1" and not available as "0" divided by the maximum value

**Table 1. Tracer the items included for measuring readiness and their definitions.**

| Items | Definitions and data sources |
|---|---|
| Guidelines on the appropriate use of blood and safe blood transfusion | Observed availability of guidelines on the appropriate use of blood and safe blood transfusion in the service area |
| Staff trained in the appropriate use of blood and safe blood transfusion | At least one staff member providing the service trained in the appropriate use of blood and safe blood transfusion within the past 2 years based on the interview response from in-charge of the service area on the day of the survey |
| Blood storage refrigerator | Observed availability and reported functionality and with temperature being monitored (checked that temperature has been monitored at least once in the past 24 hours and maintained at 2–6˚C) in the service area or adjacent area |
| Blood typing | Observed ability of the facility to conduct the ABO blood group test, Rhesus blood group test, and centrifuge tests on-site (in the facility) and functioning equipment and reagents needed to conduct the test on the day of the survey. |
| Cross-match testing | Observed ability of the facility to conduct the crossmatch (should use methods that demonstrate ABO incompatibility and incompatibility due to other clinically significant antibodies and should include an indirect anti-globulin test or a test of equivalent sensitivity), centrifuge, 37˚C incubator, and grouping sera on-site on the day of the survey. |
| Blood supply sufficiency | Reported availability of no interruption of blood availability in the last three months |
| Blood supply safety | Reported availability of blood obtained only from national or regional blood bank, OR blood obtained from other sources but screened for HIV, Syphilis, Hepatitis B, and Hepatitis C |

of 7 then multiplied by 100. A cut-off mean score of 70% or more is considered optimal for determining readiness to provide blood transfusion services [21, 22].

**Availability of basic equipment.** The availability of the following 5 items: adult weighing scale, child scale, BP apparatus, thermometer, and light source

## Data analysis

Data were analyzed with both descriptive and inferential statistics using Stata version 15. Frequency tables and summary statistics summarized facility characteristics. The chi-square ($\chi^2$) test was used to compare the relationship between managing authority and residence variables on the availability of blood transfusion services. The mean service readiness score of facilities was compared across the residence and managing authority using the Student's t-test. The one-way analysis of variance (ANOVA) F-test was performed to assess differences in the availability of blood transfusion services and facility readiness scores based on facility type and region. Furthermore, pairwise comparisons between regions and facility types were conducted, adjusting for multiple comparisons using the Bonferroni correction.

Bivariate and multivariate linear regression analyses were used to examine the predictors of blood transfusion service readiness in health facilities providing blood transfusion services. Binary linear regression was carried out to select candidate variables for the multiple regression analysis by using p-value < 0.25 as the cutoff point and theoretical framework of the study. After checking for multicollinearity using variance inflation factor (VIF), variables were entered into a multivariable linear regression model. The overall mean VIF was found to be 1.61. P-value < 0.05 was taken as the cutoff point to determine the significance of the association.

## Ethics approval and consent to participate

Ethical clearance was obtained from the Research and Ethics Committee of the Addis Continental Institute of Public Health (ACIPH). Written permission to use the data was sought from the EPHI. The researcher keeps the confidentiality and privacy of the data. Identifying information was not included in the data analysis and report.

## Results

### Facility characteristics

More facilities were from Oromia, Amhara, and SNNP regions. The majority of these facilities were health centers or clinics, located in urban centers and managed by public institutions (Table 2).

### Availability of blood transfusion services

Of the 632 facilities surveyed, 256 (41%) provided blood transfusion services. All specialized hospitals, 97% of general hospitals, and 67% of primary hospitals provided these services. As presented in Table 3 below, blood transfusion service availability significantly varied across regions, managing authority, facility type, and location.

### Service readiness in facilities offering blood transfusion services

Health facilities had a mean of 4.5 out of 7 [i.e., 65% (95% CI: 63–67%)] blood transfusion service readiness score among facilities that offer blood transfusion services and only 5% of them had all the items. Most facilities were doing blood typing but less than one-third of them were doing cross-match testing (Fig 1).

**Table 2. Sample distribution of facilities surveyed during the SARA Survey 2018.**

| | Number of facilities | Percent |
|---|---|---|
| Region | | |
| Addis Ababa | 77 | 12.2 |
| Afar | 37 | 5.9 |
| Amhara | 98 | 15.5 |
| Benishangul Gumuz | 31 | 4.9 |
| Dire Dawa | 28 | 4.4 |
| Gambella | 30 | 4.8 |
| Harari | 24 | 3.8 |
| Oromia | 109 | 17.3 |
| SNNP | 89 | 14.1 |
| Somali | 44 | 7.0 |
| Tigray | 65 | 10.3 |
| Facility type | | |
| Specialized hospital | 31 | 4.9 |
| General hospital | 116 | 18.3 |
| Primary hospital | 156 | 24.7 |
| Health center/clinic | 329 | 52.1 |
| Managing authority | | |
| Public | 410 | 64.9 |
| Private | 222 | 35.1 |
| Residence | | |
| Urban | 494 | 78.2 |
| Rural | 138 | 21.8 |
| Total | 632 | 100.0 |

Addis Ababa, Afar, and Gambella regions scored higher while Benishangul Gumuz and Somali regions had the lowest score. Availability of specific components by region, facility type, managing authority, and location are presented in Table 4 and Fig 2. The service means readiness score significantly varied across regions [F-test: 2.44; p-value: 0.009] and facility types [F-test: 5.80; p-value: < 0.001]. Pairwise comparisons revealed significant differences in the mean readiness scores between the following groups: Addis Ababa and the Somali region; specialized hospitals and primary hospitals; specialized hospitals and health centers and clinics; as well as general hospitals and health centers and clinics. However, no major difference in readiness score was observed across residence and managing authority (Table 4).

## Factors associated with blood transfusion service readiness in facilities providing blood transfusion services

The service readiness in facilities providing blood transfusion significantly varied across regions. As presented in Table 5, facilities located in Oromia (Coef.: -0.74; 95% CI: [-1.32–0.15]) and Somali (Coef.: -1.26; 95% CI: [-2.21–0.31]) had lower readiness score as compared to Addis Ababa. Facilities equipped with all basic medical equipment are positively and significantly associated with higher readiness scores. A one-unit increase in the availability of medical equipment corresponds to a 49% increase in service readiness scores (Coef.: 0.49; 95% CI: [0.19–0.79[). However, no statistically significant association between facility type, residence, or managing authority with readiness score was observed (Table 5).

**Table 3. Distributions of blood transfusion services offered by facilities according to SARA survey 2018 (N = 632).**

|  | Number of facilities offering blood transfusion | Percent | Chi² (p-value) |
|---|---|---|---|
| Region |  |  |  |
| Addis Ababa | 33 | 42.9 | **80.145 (< 0.001)** |
| Afar | 4 | 10.8 |  |
| Amhara | 56 | 57.1 |  |
| Benishangul Gumuz | 3 | 9.7 |  |
| Dire Dawa | 8 | 28.6 |  |
| Gambella | 1 | 3.3 |  |
| Harari | 5 | 20.8 |  |
| Oromia | 67 | 61.5 |  |
| SNNP | 37 | 41.6 |  |
| Somali | 16 | 36.4 |  |
| Tigray | 26 | 40.0 |  |
| Facility type |  |  |  |
| Specialized hospital | 31 | 100.0 | **435.796 < 0.001)** |
| General hospital | 112 | 96.6 |  |
| Primary hospital | 104 | 66.7 |  |
| Health center/clinic | 9 | 2.7 |  |
| Managing authority |  |  |  |
| Public | 192 | 46.8 | **19.364 (< 0.001)** |
| Private | 64 | 28.8 |  |
| Residence |  |  |  |
| Urban | 241 | 48.8 | **64.348 (< 0.001)** |
| Rural | 15 | 10.9 |  |
| Total | 256 | 40.5 |  |

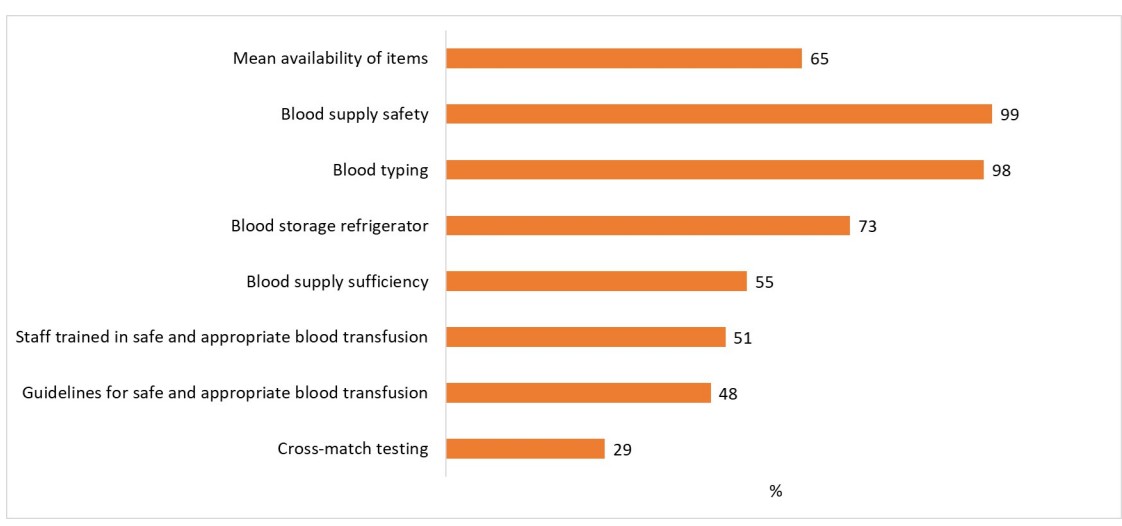

**Fig 1. Percentage of facilities that had tracer items for blood transfusion services, according to SARA 2018 (N = 256).**

**Table 4. Blood transfusion service readiness in facilities offering blood transfusion services, according to SARA survey 2018 (n = 256).**

| | Number of facilities | Guidelines available | At least 1 trained staff | Blood storage refrigerator | Blood typing | Crossmatch typing | Blood supply sufficiency | Blood supply safety | Mean number (percent) score |
|---|---|---|---|---|---|---|---|---|---|
| Region | | | | | | | | | |
| Addis Ababa | 33 | 20 (60.6) | 17 (51.5) | 28 (84.9) | 33 (100.0) | 21 (63.6) | 13 (39.4) | 33 (100.0) | 5.0 (71.4)* |
| Afar | 4 | 3 (75.0) | 2 (50.0) | 1 (25.0) | 4 (100.0) | 2 (50.0) | 4 (100.0) | 4 (100.0) | 5.0 (71.4) |
| Amhara | 56 | 27 (48.2) | 38 (67.9) | 41 (73.2) | 56 (100.0) | 14 (25.0) | 31 (55.4) | 55 (98.2) | 4.7 (66.8) |
| Benishangul Gumuz | 3 | 1 (33.3) | 2 (66.8) | 2 (66.7) | 3 (100.0) | 0 (0.0) | 0 (0.0) | 3 (100.0) | 3.7 (52.4) |
| Dire Dawa | 8 | 3 (37.5) | 6 (75.0) | 7 (87.5) | 8 (100.0) | 0 (0.0) | 4 (50.0) | 8 (100.0) | 4.5 (64.3) |
| Gambella | 1 | 1 (100.0) | 1 (100.0) | 1 (100.0) | 1 (100.0) | 0 (0.0) | 0 (0.0) | 1 (100.0) | 5.0 (71.4) |
| Harari | 5 | 3 (60.0) | 2 (40.0) | 5 (100.0) | 4 (100.0) | 2 (40.0) | 2 (40.0) | 5 (100.0) | 4.6 (65.7) |
| Oromia | 67 | 25 (37.3) | 31 (46.3) | 44 (65.7) | 64 (95.5) | 13 (19.4) | 38 (56.7) | 67 (100.0) | 4.2 (60.1) |
| SNNP | 37 | 19 (51.4) | 10 (27.0) | 31 (83.8) | 37 (100.0) | 9 (24.3) | 27 (73.0) | 37 (100.0) | 4.6 (65.6) |
| Somali | 16 | 4 (25.0) | 4 (25.0) | 7 (43.8) | 15 (93.8) | 3 (18.8) | 10 (62.5) | 15 (93.8) | 3.6 (51.8) |
| Tigray | 26 | 17 (65.4) | 17 (65.4) | 21 (80.8) | 25 (96.2) | 10 (38.5) | 11 (42.3) | 26 (100.0) | 4.9 (69.8) |
| Facility type | | | | | | | | | |
| Specialized hospital | 31 | 14 (45.2) | 19 (61.3) | 24 (77.4) | 31 (100.0) | 22 (71.0) | 16 (51.6) | 31 (100.0) | 5.1 (72.4)* |
| General hospital | 112 | 56 (50.0) | 56 (50.0) | 87 (77.7) | 108 (96.4) | 39 (34.8) | 66 (58.9) | 111 (99.1) | 4.7 (66.7) |
| Primary hospital | 104 | 51 (49.0) | 52 (50.0) | 74 (71.2) | 102 (98.1) | 11 (10.6) | 54 (51.9) | 103 (99.0) | 4.3 (61.4) |
| Health center/ clinic | 9 | 2 (22.2) | 3 (33.3) | 3 (33.3) | 9 (100.0) | 2 (22.2) | 4 (44.4) | 9 (100.0) | 3.6 (50.8) |
| Managing authority | | | | | | | | | |
| Public | 192 | 93 (48.4) | 101 (52.6) | 138 (71.9) | 186 (96.9) | 46 (24.0) | 112 (58.3) | 190 (97.0) | 4.5 (64.4) |
| Private | 64 | 30 (46.9) | 29 (45.3) | 50 (78.1) | 64 (100.0) | 28 (43.8) | 28 (43.8) | 64 (100.0) | 4.6 (65.4) |
| Residence | | | | | | | | | |
| Urban | 241 | 117 (48.6) | 123 (51.0) | 179 (74.3) | 235 (97.5) | 71 (29.5) | 132 (54.8) | 239 (99.2) | 4.5 (65.0) |
| Rural | 15 | 6 (40.0) | 7 (46.7) | 9 (60.0) | 15 (100.0) | 3 (20.0) | 8 (53.3) | 15 (100.0) | 4.2 (60.0) |

*p-value <0.01

## Discussion

This study identified limited availability of blood transfusion services and suboptimal readiness in the facilities that provide these services. Furthermore, the readiness score varied significantly across regions.

The study revealed that a significant number of hospitals and most health centers and clinics lacked blood transfusion services, despite the expectation that hospitals and certain upgraded health centers and clinics should offer these services—particularly for emergencies, maternal health care, and other critical care situations. As the SARA survey provided a conservative estimate of the all-time availability of blood transfusion services, highlighting a concerning situation that would significantly impact access to emergency care. Previous studies have

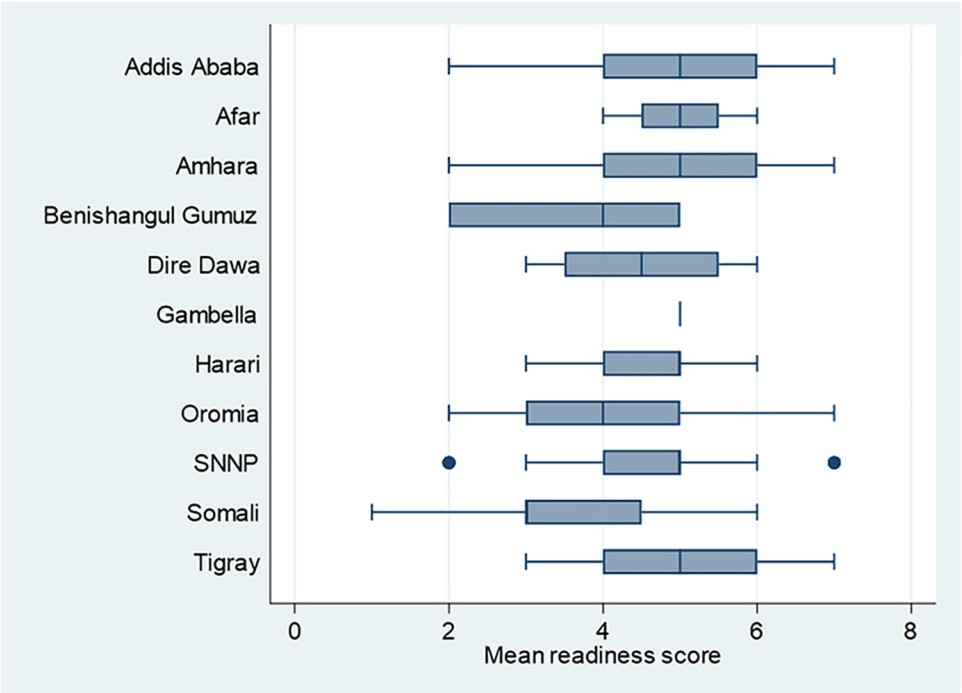

**Fig 2. Mean service readiness score in facilities offering blood transfusion services, by region, according to SARA survey 2018 (n = 256).**

also shown that limited access, primarily due to the physical distance to facilities, is a major factor contributing to delays in receiving necessary care [23–25].

The readiness of health facilities to provide safe blood transfusion service is a critical aspect of healthcare delivery, impacting patient outcomes and safety. However, the assessment of readiness in facilities providing blood transfusion services was suboptimal. This indicates room for improvement in ensuring that essential requirements for safe and effective blood transfusion services are delivered across all facilities. The national Emergency Obstetric and Newborn Care (EmONC) [26] and Service Provision Assessment (SPA) [27] assessments also showed that less than two-thirds of hospitals or maternal and child health specialty centers were ready to provide blood transfusion services for obstetric indications in the last 3 months before the study. Likewise, in eight African countries, the readiness of facilities to provide blood transfusion services for obstetric indications ranged from 54–80% [28].

In this study, most facilities offering blood transfusion services were not doing cross-match testing and more than half of the facilities lacked guidelines and trained staff. Previous studies also reported challenges in the availability of basic amenities, medical equipment, adherence to infection prevention precautions, and sufficient technical staff were key determinants of facility readiness [28, 29]. Facilities equipped with these essential components demonstrated higher readiness scores, highlighting the critical role of infrastructure, resources, and staffing in delivering quality transfusion services. The study also identified a positive association between the availability of medical equipment and service readiness. This underscores the importance of investing in appropriate equipment and ensuring consistent supply to enable facilities to deliver quality and safe blood transfusion services.

Regionally, facilities in Oromia and the Somali regions exhibited lower readiness scores. This suggests specific challenges or resource constraints in these regions that need to be

**Table 5. Factors affecting service readiness in facilities offering blood transfusion services, according to SARA survey 2018 (n = 256).**

| | Number of facilities | Unadjusted | | | Adjusted | | |
|---|---|---|---|---|---|---|---|
| | | Coef. | 95% CI | P-value | Coef. | 95% CI | P-value |
| Region | | | | | | | |
| Addis Ababa (ref) | 33 | 1.00 | | | 1.00 | | |
| Afar | 4 | -2.00E-16 | -1.25–1.25 | 1.000 | 0.003 | -0.86–0.87 | 0.995 |
| Amhara | 56 | -0.32 | -0.84–0.2 | 0.225 | -0.22 | -0.84–0.40 | 0.487 |
| Benishangul Gumuz | 3 | -1.33 | -2.76–0.1 | 0.067 | -1.18 | -3.15–0.79 | 0.240 |
| Dire Dawa | 8 | -0.50 | -1.43–0.43 | 0.293 | -0.31 | -1.16–0.54 | 0.476 |
| Gambella | 1 | -1.00E-16 | -2.4–2.4 | 1.000 | -0.18 | -0.73–0.37 | 0.524 |
| Harari | 5 | -0.40 | -1.54–0.74 | 0.489 | -0.36 | -1.42–0.69 | 0.498 |
| Oromia | 67 | **-0.80** | **-1.29–0.29** | **0.002** | **-0.74** | **-1.32–0.15** | **0.014** |
| SNNP | 37 | -0.41 | -0.97–0.16 | 0.161 | -0.29 | -0.96–0.38 | 0.392 |
| Somali | 16 | **-1.34** | **-2.1–0.65** | **< 0.001** | **-1.26** | **-2.21–0.31** | **0.009** |
| Tigray | 26 | -0.12 | -0.74–0.51 | 0.715 | 0.05 | -0.66–0.75 | 0.894 |
| Facility type | | | | | | | |
| Hospital | 247 | 1.01 | **0.19–1.83** | **0.016** | 0.28 | -0.83–1.38 | 0.623 |
| Health center/clinic (ref) | 9 | 1.00 | | | 1.00 | | |
| Managing authority | | | | | | | |
| Public | 192 | -0.07 | -0.42–0.28) | 0.705 | 0.05 | -0.40–0.50 | 0.833 |
| Private (ref) | 64 | 1.00 | | | 1.00 | | |
| Residence | | | | | | | |
| Urban | 241 | 0.35 | -0.30–1.00 | 0.291 | 0.23 | -0.35–0.82 | 0.430 |
| Rural (ref) | 15 | 1.00 | | | 1.00 | | |
| Availability of medical equipment | 256 | 0.44 | **0.004 (0.14–0.73)** | | 0.49 | **0.19–0.79** | **0.001** |
| _cons | | 4.29 | 4.09–4.50 | < 0.001 | 4.13 | 2.79–5.47 | < 0.001 |

addressed to enhance facility readiness for blood transfusion services. Strategies tailored to the unique needs and constraints of each region could be pivotal in improving transfusion service readiness and ultimately healthcare outcomes. The national EmONC [26] and SPA [27] assessments also showed regional variations in the readiness to provide blood transfusion services. The study of eight African countries also showed variations in the readiness of facilities providing blood transfusion services across countries [28].

Using nationally representative data, this study provides comprehensive insights into health facilities' readiness to offer blood transfusion services in Ethiopia. The findings from this study provide valuable information that can inform targeted interventions and resource allocation strategies aimed at enhancing the safety and quality of blood transfusion services in Ethiopia. However, it has limitations. The study's limitations stem primarily from its reliance on secondary data, which constrained the inclusion of important factors such as patient load, incentive mechanisms, and characteristics of facility managers that could influence blood transfusion service readiness in Ethiopia. Another limitation is time-lag bias. The static nature of secondary data may not fully capture dynamic and contextual factors that evolve within the healthcare system, highlighting the need for more nuanced and real-time data sources to better understand the complex determinants of facility readiness in this critical area of healthcare delivery.

## Conclusion

The study identified suboptimal blood transfusion service readiness and significant regional variations. Availability of medical equipment and administrative regions were significantly

associated with blood transfusion service readiness. To enhance facility readiness, collaborative efforts are needed to invest in the necessary medical equipment and address regional variations. It is imperative to provide tailored and demand-driven assistance to regions, facilitating the upgrading of their facilities to meet the requisite standards of readiness.

## Author Contributions

**Conceptualization:** Tirualem Asmare Fenta, Gizachew Tadele Tiruneh, Nebiyou Fasil Ayele.

**Data curation:** Tirualem Asmare Fenta, Gizachew Tadele Tiruneh.

**Formal analysis:** Tirualem Asmare Fenta, Gizachew Tadele Tiruneh, Nebiyou Fasil Ayele.

**Investigation:** Tirualem Asmare Fenta.

**Methodology:** Tirualem Asmare Fenta, Gizachew Tadele Tiruneh, Nebiyou Fasil Ayele.

**Project administration:** Tirualem Asmare Fenta, Nebiyou Fasil Ayele.

**Software:** Tirualem Asmare Fenta.

**Supervision:** Nebiyou Fasil Ayele.

**Validation:** Nebiyou Fasil Ayele.

**Visualization:** Tirualem Asmare Fenta, Gizachew Tadele Tiruneh, Nebiyou Fasil Ayele.

**Writing – original draft:** Tirualem Asmare Fenta, Gizachew Tadele Tiruneh, Nebiyou Fasil Ayele.

**Writing – review & editing:** Tirualem Asmare Fenta, Gizachew Tadele Tiruneh, Nebiyou Fasil Ayele.

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
