## [Decision Letter · Decision Letter 0]

20 Sep 2024

PONE-D-24-30780Blood transfusion service readiness and its associated factors in health facilities providing blood transfusion services across Ethiopia: A secondary analysis of the 2018 SARA surveyPLOS ONE

Dear Dr. Tiruneh,

Thank you for submitting your manuscript to PLOS ONE. After careful consideration, we feel that it has merit but does not fully meet PLOS ONE’s publication criteria as it currently stands. Therefore, we invite you to submit a revised version of the manuscript that addresses the points raised during the review process.

 Please make the requested alterations outlined below prior to further review.

We look forward to receiving your revised manuscript.

Kind regards,

Rena Hirani

Academic Editor

PLOS ONE

Journal Requirements:

2. For studies involving third-party data, we encourage authors to share any data specific to their analyses that they can legally distribute. PLOS recognizes, however, that authors may be using third-party data they do not have the rights to share. When third-party data cannot be publicly shared, authors must provide all information necessary for interested researchers to apply to gain access to the data. (https://journals.plos.org/plosone/s/data-availability#loc-acceptable-data-access-restrictions)  

3. Please upload a new copy of Figure 1 as the detail is not clear. Please follow the link for more information:

https://blogs.plos.org/plos/2019/06/looking-good-tips-for-creating-your-plos-figures-graphics/

https://blogs.plos.org/plos/2019/06/looking-good-tips-for-creating-your-plos-figures-graphics/

4. Please include a copy of Table 5 which you refer to in your text on page 13.

**Additional Editor Comments:**

Please define the terms SSA and SARA as requested.

Please take note of the statistical terminology requested by reviewers.

Reviewers' comments:

Reviewer's Responses to Questions

**Comments to the Author**

1. Is the manuscript technically sound, and do the data support the conclusions?

Reviewer #1: Partly

Reviewer #2: Yes

Reviewer #3: Yes

2. Has the statistical analysis been performed appropriately and rigorously? 

Reviewer #1: I Don't Know

Reviewer #2: Yes

Reviewer #3: Yes

3. Have the authors made all data underlying the findings in their manuscript fully available?

Reviewer #1: Yes

Reviewer #2: Yes

Reviewer #3: Yes

4. Is the manuscript presented in an intelligible fashion and written in standard English?

Reviewer #1: Yes

Reviewer #2: Yes

Reviewer #3: Yes

5. Review Comments to the Author

Reviewer #1: Thanks to the authors for this interesting manuscript. I have a few suggested edits/comments that you may wish to consider. Critically, it would be helpful to have a bit more methodological clarity and detail to enable a full interpretation of results.

Abstract

- What is meant by “determinants” of blood transfusion services? I think keeping it to readiness alone better reflects what was carried out.

- Please swap commas for semi-colons between the components in lines 19–22.

Introduction

- In the first two sentences, there is no link made to the importance of blood transfusion for these conditions.

- It’s also not clear in lines 41/41 (over half of the deaths result from conditions that can be managed through prehospital and facility-based emergency care) what the reference is to needing blood. Please make this more explicit.

- Please define SSA (line 50)

- Before you introduce what influences readiness, it would be worthwhile to define what you mean by readiness within the context of this study

- Remove “the” in line 58

- Please remove any acronyms that are not repeated in the manuscript (e.g. ERCS)

Methods

- It’s not clear how many facilities participated at what level in your study. The 2018 SARA included 764 facilities, a census of 303 hospitals, 164 health centers, 165 clinics, excluding the health posts, but you note that you included 632 facilities, excluding health posts, participated in your study? Why not all of the facilities in the SARA? It would also be helpful to see what those 632 facilities were in terms of level. Please clarify what the overall sample size (and if there was a calculation, please provide this) was and how it was expected to be generalizable (i.e. what parameters you used for sample size calculations to ensure this, especially as regressions were used—how did you ensure you had sufficient numbers for each variable to do this type of analysis?).

- To that end, a table would be enormously useful (e.g. with a column for total health facilities by region/type, a column for those included in the SARA, and then a column for those included in this analysis).

- Please remove “were” in line 96

- Interviews and a facility inventory questionnaire were carried out for the SARA, but it’s not clear what the function of the interviews was in terms of readiness? Which instrument actually gave insights around the seven tracer items used in the present study?

- Please describe the seven tracer items in more detail, in terms of what was actually assessed for each, the time frames in which they may have been evaluated (e.g. previous three months? Only the day of the assessment?)

- Please add “and” before “service readiness” in line 113

Results

- Please add a “totals” row in Table 1.

- The chi-squared test is not mentioned in your methods. Further, it’s not clear to me why this was done at all? For instance, would you expect all facilities to provide blood transfusion? In Table 2, please ensure n/N is reflected. It’s not clear what the total facilities is (and further, is this ALL facilities in the region, or only those expected to carry out blood transfusion? If it’s the latter, the chi-squared is useful, but if it’s the former, I would not include this analysis).

- This is a minor comment, but please swap the [ ] and ( ) brackets (e.g. (Coef.: -0.74; 95% CI: 0.014 [-1.32--0.15]). In general, confidence intervals are usually in square brackets.

Discussion

- In line 192, please change “was significantly varied” to “varied significantly”.

- Please revise “the readiness in facilities providing” in line 201 to “the readiness of facilities to provide”.

- A limitation of this study is that it relies on the SARA, which only allowed for “available” or “not available” , but I imagine in many facilities (as has been my own experience), there is availability “sometimes”. How were these facilities coded? If it had to be all the time to be coded “available”, then the overall picture presented would be quite conservative, reflecting a worse picture. If there just had to be some availability for “available”, then the converse is true. It would be useful for the authors to reflect on the implications of this coding for reflecting the overall state of blood transfusion services.

- It also strikes me as a clear finding that only 256 facilities actually provide blood transfusion, in practice, as I imagine the number, in theory is much higher. In your discussion you might consider adding a reflection on the proportion of facilities expected to offer blood transfusion services that actually do and the implications of this.

- It would also be useful to see some reflections by the authors on the implications of this study for both policy and for practice.

- Please revise “is” to “were” in line 235, and “investments” to “investing”, “in availing” to “to avail”, and “addressing” to “to address” in line 237.

- In the list of acronyms, it should be “odds ratio”, not “odd ratio”

Reviewer #2: A research study was conducted which aimed to examine the readiness and determinants of blood transfusion services in Ethiopian facilities.

Minor revisions:

1- The standard statistical term for average is mean. When providing means, also state the corresponding standard deviations.

2- Line 127: To improve clarity consider replacing “for” to “with”.

3- Line 136: Add “for” to “check for multicollinearity”.

4- P-values never equal zero; express small p-values as < 0.001.

5- Line 169: State which pairwise regions and facility types differed with respect to mean readiness scores.

6- Line 128: The Student’s t-test is appropriate for comparing two groups. Locations and facility types have more than two groups, correct? ANOVA is an extension of the Student's t-test for comparing more than two groups.

Reviewer #3: Methods

- Measurement: I am not clear on how you obtain the mean percentage service readiness score. The description seemed confusing to me. What are the cut-off points?

Results

- Please do not represent # as number. It is not a standard symbol in journals

- Line 152, first sentence is a repetition from your methods. I suggest rewording them

- p-value of 0.000 should be changed to <0.001

- Line 179: Table 5 does not exist

- Please change the way you present your regression analysis. Your p-value and 95%CI are reversed

Discussion

- How do you determine your results to be sub-optimal or optimal? Refer to methods comment.

- We need more comparisons with other studies. If not African countries, how about other low-middle income countries for comparison?

- Discussions for the availability of medical equipment is lacking. Please expand on your findings.

Miscellaneousc

- Please spell out the full term at its first mention, indicate its abbreviation in parenthesis and use the abbreviation from then on (SSA, SARA)

6. PLOS authors have the option to publish the peer review history of their article (what does this mean?). If published, this will include your full peer review and any attached files.

Reviewer #1: No

Reviewer #2: No

Reviewer #3: No

---

## [Author Response · Author response to Decision Letter 0]

25 Sep 2024

A point-by-point response to reviews

Dear Editor,

We, the authors, would like to thank the reviewers for their valuable comments which we feel have significantly strengthened our paper. Our point-by-point responses to the reviewers and editor are below each comment. We also reviewed to ensure that this version of the manuscript conforms to the journal style.

Response to Reviewers

Journal Requirements:

Response: In this version, we have updated the figure in-text citations accordingly and ensured that the manuscript adheres to PLOS ONE's style requirements.

2. For studies involving third-party data, we encourage authors to share any data specific to their analyses that they can legally distribute. PLOS recognizes, however, that authors may be using third-party data they do not have the rights to share. When third-party data cannot be publicly shared, authors must provide all information necessary for interested researchers to apply to gain access to the data. (https://journals.plos.org/plosone/s/data-availability#loc-acceptable-data-access-restrictions) 

Response: We updated the Data Availability Statement as follows: 

Conducted by the Ethiopian Public Health Institute (EPHI) between October and December 2017, the SARA 2018 dataset—generated from a facility-based cross-sectional survey—offers insights into service availability and readiness for general healthcare and various specific services at both national and regional levels. Interested researchers can access the dataset by following a data-sharing agreement.

Researchers can request the dataset through EPHI’s data repository website for the “Ethiopia Services Availability and Readiness Assessment (SARA) Survey, 2018,” available at https://rtds.ephi.gov.et/public/showdetail/94. An assigned officer will review and facilitate the request. Upon submission, researchers will receive an email outlining the requirements, which include: 1) completing and returning the data request form; 2) providing a copy of the ethical approval certificate; and 3) attaching an official data request letter.

3. Please upload a new copy of Figure 1 as the detail is not clear. Please follow the link for more information:

https://blogs.plos.org/plos/2019/06/looking-good-tips-for-creating-your-plos-figures-graphics/

https://blogs.plos.org/plos/2019/06/looking-good-tips-for-creating-your-plos-figures-graphics/

Response: Figure 1 with improved resolution has been uploaded.

4. Please include a copy of Table 5 which you refer to in your text on page 13.

Response: It was a typo referring to Table 4, which has now been corrected. 

Response: Thank you. The references are checked and corrected for completeness. 

Additional Editor Comments:

Please define the terms SSA and SARA as requested.

Please take note of the statistical terminology requested by reviewers.

Response: Thank you. The abbreviations have been defined, and all necessary changes have been made according to the reviewers' comments. 

Reviewers' comments:

Reviewer #1: Thanks to the authors for this interesting manuscript. I have a few suggested edits/comments that you may wish to consider. Critically, it would be helpful to have a bit more methodological clarity and detail to enable a full interpretation of results.

Response: We well-acknowledged the valuable comments. In this version, we have addressed them and improved the manuscript accordingly.

Abstract

- What is meant by “determinants” of blood transfusion services? I think keeping it to readiness alone better reflects what was carried out.

Response: Thank you for the valid comments. Comment well taken and edited as follows “This study examined the readiness of blood transfusion services in health facilities across Ethiopia and its determinants.”

- Please swap commas for semi-colons between the components in lines 19–22.

Response: Comment well taken and edited accordingly.

Introduction

- In the first two sentences, there is no link made to the importance of blood transfusion for these conditions.

- It’s also not clear in lines 41/41 (over half of the deaths result from conditions that can be managed through prehospital and facility-based emergency care) what the reference is to needing blood. Please make this more explicit.

Response: The texts of the first two sentences as well as lines 41/42 have been revised for clarity as follows “Safe blood transfusion is essential for treating injuries and emergency obstetric conditions, serving as a life-saving intervention in various clinical scenarios. Timely and secure blood transfusion services are particularly vital in LMICs, where over half of deaths result from conditions manageable through emergency care.”

- Before you introduce what influences readiness, it would be worthwhile to define what you mean by readiness within the context of this study

Response: Thank you for the valid comments. The following definition is included in the text “Blood transfusion service readiness —defined as the preparedness and capability of a facility to provide safe, timely, and effective blood products and services, which includes trained staff, blood storage refrigerators, blood typing, cross-matching, supply safety and sufficiency, and guidelines for appropriate use—is crucial for ensuring safe and effective transfusions [12].” Lines 55-58

- Please define SSA (line 50)

- Remove “the” in line 58

- Please remove any acronyms that are not repeated in the manuscript (e.g. ERCS)

Response: We have defined the acronym SSA, removed 'the' where necessary, and eliminated the acronym ERCS accordingly.

Methods

- It’s not clear how many facilities participated at what level in your study. The 2018 SARA included 764 facilities, a census of 303 hospitals, 164 health centers, 165 clinics, excluding the health posts, but you note that you included 632 facilities, excluding health posts, participated in your study? Why not all of the facilities in the SARA? It would also be helpful to see what those 632 facilities were in terms of level. Please clarify what the overall sample size (and if there was a calculation, please provide this) was and how it was expected to be generalizable (i.e. what parameters you used for sample size calculations to ensure this, especially as regressions were used—how did you ensure you had sufficient numbers for each variable to do this type of analysis?).

- To that end, a table would be enormously useful (e.g. with a column for total health facilities by region/type, a column for those included in the SARA, and then a column for those included in this analysis).

Response: The 2018 SARA included 632 hospitals, health centers, and clinics, along with 162 health posts. We included all 632 facilities (excluding the 132 health posts, which are not expected to provide blood transfusion services). Tables 1 and 2 present the facility levels and location. Regarding the adequacy of the sample size check, we have added the following sentences “In this study, we included all 632 facilities (except the health posts which are expected to provide blood transfusion services) in the SARA survey were included. The adequacy of the sample size is assessed using a single proportion formula with a 5% margin of error, a 95% confidence level, a design effect of 1.5, and an assumption that 88% of hospitals reported performing compatibility testing for whole blood and RBC [20]. Additionally, a double population comparison is conducted for blood transfusion service readiness, considering managing authority and facility type as exposure variables, with an assumed power of 80%, a design effect of 1.5, and a 95% confidence interval [16].” Lines 104-109 

- Please remove “were” in line 96

- Please add “and” before “service readiness” in line 113

Response: We have removed the word 'were' and added 'and' accordingly.

- Interviews and a facility inventory questionnaire were carried out for the SARA, but it’s not clear what the function of the interviews was in terms of readiness? Which instrument actually gave insights around the seven tracer items used in the present study?

Response: Thank you for the comments. The following sentence has been added in lines 120-122. “Specific to facility readiness, the interviewer was instructed to interview the person most knowledgeable about blood transfusion services at the facility where blood is collected, processed, tested, stored, or handled prior to transfusion.”

- Please describe the seven tracer items in more detail, in terms of what was actually assessed for each, the time frames in which they may have been evaluated (e.g. previous three months? Only the day of the assessment

Response: Thanks for the valid comments. The table below has been added to clarify each tracer items detailing what it means including its time timeframe. Lines 128-139

Items Definitions and data sources

Guidelines on the appropriate use of blood and safe blood transfusion Observed availability of guidelines on the appropriate use of blood and safe blood transfusion in the service area

Staff trained in the appropriate use of blood and safe blood transfusion At least one staff member providing the service trained in the appropriate use of blood and safe blood transfusion within the past 2 years based on the interview response from in-charge of service area day of the survey

Blood storage refrigerator Observed availability and reported functionality and with temperature being monitored (checked that temperature has been monitored at least once in the past 24 hours and maintained at 2 – 6 oC) in the service area or adjacent area

Blood typing Observed ability of the facility to conduct the ABO blood group test, Rhesus blood group test, and centrifuge tests on-site (in the facility) and functioning equipment and reagents needed to conduct the test on the day of the survey. 

Cross-match testing Observed ability of the facility to conduct the crossmatch (should use methods that demonstrate ABO incompatibility and incompatibility due to other clinically significant antibodies and should include an indirect anti-globulin test or a test of equivalent sensitivity), centrifuge, 37°C incubator, and grouping sera on-site on the day of the survey.

Blood supply sufficiency Reported availability of no interruption of blood availability in the last three months

Blood supply safety Reported availability of blood obtained only from national or regional blood bank, OR blood obtained from other sources but screened for HIV, Syphilis, Hepatitis B, and Hepatitis C

Results

- Please add a “totals” row in Table 1.

- The chi-squared test is not mentioned in your methods. Further, it’s not clear to me why this was done at all? For instance, would you expect all facilities to provide blood transfusion? In Table 2, please ensure n/N is reflected. It’s not clear what the total facilities is (and further, is this ALL facilities in the region, or only those expected to carry out blood transfusion? If it’s the latter, the chi-squared is useful, but if it’s the former, I would not include this analysis).

Response: The total number of facilities expected to provide blood transfusion services are 632 sampled facilities in the country in the SARA survey. The N and n/N are included in Tables 1 and 2, respectively. Ch2 is mentioned in the analysis section. Lines 147-149.

- This is a minor comment, but please swap the [ ] and ( ) brackets (e.g. (Coef.: -0.74; 95% CI: 0.014 [-1.32--0.15]). In general, confidence intervals are usually in square brackets.

Discussion

- In line 192, please change “was significantly varied” to “varied significantly”.

- Please revise “the readiness in facilities providing” in line 201 to “the readiness of facilities to provide”.

Response: Thank you for bringing all these to our attention. These have been edited accordingly.

- A limitation of this study is that it relies on the SARA, which only allowed for “available” or “not available” , but I imagine in many facilities (as has been my own experience), there is availability “sometimes”. How were these facilities coded? If it had to be all the time to be coded “available”, then the overall picture presented would be quite conservative, reflecting a worse picture. If there just had to be some availability for “available”, then the converse is true. It would be useful for the authors to reflect on the implications of this coding for reflecting the overall state of blood transfusion services.

Response: This is an interesting insight. The survey estimates the all-time availability of blood transfusion services, which is indeed a concerning situation as you noted. We have reflected this in the Discussion section, lines 226-228. “As the SARA survey provided a conservative estimate of the all-time availability of blood transfusion services, highlighting a concerning situation that would significantly impact access to emergency care. “

- It also strikes me as a clear finding that only 256 facilities actually provide blood transfusion, in practice, as I imagine the number, in theory is much higher. In your discussion you might consider adding a reflection on the proportion of facilities expected to offer blood transfusion services that actually do and the implications of this.

- It would also be useful to see some reflections by the authors on the implications of this study for both policy and for practice.

Response: Comment well taken. We have discussed this dedicating a paragraph Lines 223-230 

“The study revealed that a significant number of hospitals and most health centers and clinics lacked blood transfusion services, despite the expectation that hospitals and certain upgraded health centers and clinics should offer these services—particularly for emergencies, maternal health care, and other critical care situations. As the SARA survey provided a conservative estimate of the all-time availability of blood transfusion services, highlighting a concerning situation that would significantly impact access to emergency care. Previous studies have also shown that limited access, primarily due to the physical distance to facilities, is a major factor contributing to delays in receiving necessary care [21-23].”

- Please revise “is” to “were” in line 235, and “investments” to “investing”, “in availing” to “to avail”, and “addressing” to “to address” in line 237.

- In the list of acronyms, it should be “odds ratio”, not “odd ratio”

Response: Thank you again for bringing them to our attention. We have edited them accordingly. 

Reviewer #2: A research study was conducted which aimed to examine the readiness and determinants of blood transfusion services in Ethiopian facilities.

Minor revisions:

1- The standard statistical term for average is mean. When prov

---

## [Decision Letter · Decision Letter 1]

26 Nov 2024

PONE-D-24-30780R1Blood transfusion service readiness and its associated factors in health facilities providing blood transfusion services across Ethiopia: A secondary analysis of the 2018 Service Availability and Readiness Assessment (SARA) surveyPLOS ONE

Dear Dr. Tiruneh,

Thank you for submitting your manuscript to PLOS ONE. After careful consideration, we feel that it has merit but does not fully meet PLOS ONE’s publication criteria as it currently stands. Therefore, we invite you to submit a revised version of the manuscript that addresses the points raised during the review process.

We look forward to receiving your revised manuscript.

Kind regards,

Alemu Birara Zemariam

Academic Editor

PLOS ONE

**Journal Requirements:**

Reviewers' comments:

Reviewer's Responses to Questions

**Comments to the Author**

1. If the authors have adequately addressed your comments raised in a previous round of review and you feel that this manuscript is now acceptable for publication, you may indicate that here to bypass the “Comments to the Author” section, enter your conflict of interest statement in the “Confidential to Editor” section, and submit your "Accept" recommendation.

Reviewer #1: All comments have been addressed

Reviewer #2: (No Response)

Reviewer #3: (No Response)

2. Is the manuscript technically sound, and do the data support the conclusions?

Reviewer #1: Yes

Reviewer #2: Yes

Reviewer #3: Yes

3. Has the statistical analysis been performed appropriately and rigorously? 

Reviewer #1: I Don't Know

Reviewer #2: Yes

Reviewer #3: Yes

4. Have the authors made all data underlying the findings in their manuscript fully available?

Reviewer #1: Yes

Reviewer #2: Yes

Reviewer #3: Yes

5. Is the manuscript presented in an intelligible fashion and written in standard English?

Reviewer #1: Yes

Reviewer #2: Yes

Reviewer #3: Yes

6. Review Comments to the Author

**Reviewer #1:** Many thanks to the authors for their revised manuscript. This is reading more clearly now and I'm sure will make a useful contribution to literature.

**Reviewer #2: **Minor revision:

Line 104: For the sample size estimate, indicate the statistical testing method used to generate the confidence interval.

**Reviewer #3: **Thank you for addressing all the reviewers' comments in your manuscript. I still have one issue with the writing that has not been resolved.

In your methods, you have clearly outlined the seven tracer items and their definitions, and mentioned how you obtain the mean percentage service readiness score. Now, how do you determine what is the cut-off mean score?

Situation 1:

If 7 is the maximum score, would a mean score of 3.5 (50%) be the cut-off point between being optimum (>50%) and suboptimum (<50%)?

Situation 2:

A 70% a cut-off point, according to other SARA studies I've found.

Situation 3:

You simply categorize any score <100% a suboptimal score, according to what you have described in your discussion. In table 4, all mean scores according to the region, facility, managing authority and residence is at the range of 50% - 72.4%. Therefore, the service readiness is at "suboptimal" level?

Whether the mean score readiness is set arbitrarily, or at a standard, or otherwise, please mention it in your methods.

7. PLOS authors have the option to publish the peer review history of their article (what does this mean?). If published, this will include your full peer review and any attached files.

Reviewer #1: No

Reviewer #2: No

Reviewer #3: No

---

## [Author Response · Author response to Decision Letter 1]

27 Nov 2024

A point-by-point response to reviews

Dear Alemu Birara, 

Academic Editor, 

We, the authors, would like to express our gratitude to the reviewers and editors of PLOS ONE Journal for their valuable comments, which we believe have significantly strengthened our paper. Below, we provide point-by-point responses to each comment and suggestions in red font. We have also reviewed the manuscript to ensure that it conforms to the journal's style.

Journal Requirements

Response: The reference lists have been reviewed, and minor corrections have been made to the format

Reviewers' comments

Reviewer 1

All comments have been addressed. Many thanks to the authors for their revised manuscript. This is reading more clearly now and I'm sure will make a useful contribution to literature.

Response: Thank you for taking the time to review our manuscript.

Reviewer 2

Minor revision:

Line 104: For the sample size estimate, indicate the statistical testing method used to generate the confidence interval.

Response: Thank you for bringing this to our attention. We used the Wilson score interval method to generate the confidence interval for the proportion estimate and to assess the adequacy of the sample size. The following sentence has been included in the revised version “The Wilson score interval method was used to generate the confidence interval for the proportion estimate." See the Revised Manuscript with Track Changes, page 7, lines 105–106. 

Thank you for your valuable feedback.

Reviewer 3

Thank you for addressing all the reviewers' comments in your manuscript. I still have one issue with the writing that has not been resolved.

In your methods, you have clearly outlined the seven tracer items and their definitions, and mentioned how you obtain the mean percentage service readiness score. Now, how do you determine what is the cut-off mean score?

Situation 1:

If 7 is the maximum score, would a mean score of 3.5 (50%) be the cut-off point between being optimum (>50%) and suboptimum (<50%)?

Situation 2:

A 70% a cut-off point, according to other SARA studies I've found.

Situation 3:

You simply categorize any score <100% a suboptimal score, according to what you have described in your discussion. In table 4, all mean scores according to the region, facility, managing authority and residence is at the range of 50% - 72.4%. Therefore, the service readiness is at "suboptimal" level?

Whether the mean score readiness is set arbitrarily, or at a standard, or otherwise, please mention it in your methods.

Response: Thank you for your insightful explanation and guidance. We sincerely apologize for overlooking your comments in the previous version. Initially, we believed we had used the mean readiness score as a continuous variable. However, as you rightly pointed out, we categorized it into suboptimal and optimal scores in our Discussion section, while also reporting the mean score of 65% in the Results section. 

Accordingly, we have applied the 70% cut-off point, as you suggested, and as is commonly used in previous SARA-based surveys for assessing readiness in chronic disease services. The following sentence has been included in the revised version, with the corresponding references cited: "A cut-off mean score of 70% or more is considered optimal for determining readiness to provide blood transfusion services [21, 22].", Revised Manuscript with Track Changes, page 9, lines 142–143. 

Thank you for your thorough review and guidance.

---

## [Editor Report · Decision Letter 2]

29 Nov 2024

Blood transfusion service readiness and its associated factors in health facilities providing blood transfusion services across Ethiopia: A secondary analysis of the 2018 Service Availability and Readiness Assessment (SARA) survey

PONE-D-24-30780R2

Dear Dr. Tiruneh,

We’re pleased to inform you that your manuscript has been judged scientifically suitable for publication and will be formally accepted for publication once it meets all outstanding technical requirements.

Kind regards,

Alemu Birara Zemariam

Academic Editor

PLOS ONE

---

## [Editor Report · Acceptance letter]

3 Dec 2024

PONE-D-24-30780R2 

PLOS ONE

Dear Dr. Tiruneh, 

I'm pleased to inform you that your manuscript has been deemed suitable for publication in PLOS ONE. Congratulations! Your manuscript is now being handed over to our production team.

Kind regards, 

on behalf of

Dr Alemu Birara Zemariam 

Academic Editor

PLOS ONE